# Economic effects of conversion from county (or county-level city) to municipal district in China

**Biao Zhao**[1]*, **Xu Xi**[2]

**1** Institute of Chinese Borderland Studies, CASS, Beijing, China, **2** School of Geography Science and Geomatics Engineering, Suzhou University of Science and Technology, Suzhou, Jiangsu, China

* zhaobiao@igsnrr.ac.cn

**Data Availability Statement:** All relevant data are within the paper. The "Positivism Model and Data Source" in the paper describes the data source and processing process. The original data of the paper

## Abstract

Administrative division is an important resource to promote the urbanization process and economic growth in China. As an important way of urban spatial expansion, the effect of the removal of counties (county-level cities) into municipal districts(RCD) on economic growth remains to be empirically tested. In this paper, the panel data at the county level from 1998 to 2016 and the differential method were selected to study this problem. The results show that, during the study period, the RCD significantly promoted the economic growth of Chinese cities. The effect of removing counties (county-level cities) from large cities and mega-cities to set up districts is obviously better than that of small and medium-sized cities. In small and medium-sized cities with small urban permanent population, the RCD has obvious negative impact on economic development. The effect of county (county-level city) reform in eastern and central regions is more significant, while the effect of policy in western and northeast regions is not significant. When the development intensity of the municipal district is between 15%-20%, the effect of the RCD is relatively good, and the administrative division adjustment of the municipal district has a certain optimal window period.

## Introduction

Administrative division is the spatial foundation of national and urban governance, and an important means of political power construction and management, resource integration and distribution. The scientific and reasonable establishment of administrative divisions is directly related to the efficiency of administrative management and the stability of political power, so it is widely valued by all countries in the world [1–4]. The adjustment of administrative divisions is the objective need to adapt to economic and social development in different urbanization periods. During the period of rapid urbanization, developed countries such as Britain, Germany, Japan and South Korea have also carried out significant adjustment or reform of administrative divisions according to their urban regional functions, resource allocation and administrative needs. Since the reform and opening up, China is experiencing the largest and fastest urbanization process in the world history. It is generally believed that the optimization

are in attachment 1. Data relevant to the adjustment of administrative division are derived from the China Administrative Division Network (http://www.xzqh.org/html/). The economic data of cities come from The Statistical Yearbook of Chinese Cities and Statistical Yearbook of Chinese Urban Construction provided by China Economic and Social Big Data Research Platform (https://data.cnki.net/Yearbook/Navi?type=type&code=A) in past years, and the data of the counties (cities) removed from the merger come from the Statistical Yearbook of Chinese Counties and the statistical yearbook of provinces and cities in past years.

**Funding:** The authors acknowledge the funding received from the National Natural Science Foundation of China to conduct this study (Grant No.42001129).

**Competing interests:** The authors have declared that no competing interests exist.

of administrative divisions plays an important role in the improvement of China's urbanization level [5, 6]. In current China, however, trans-regional reconfiguration of factors of production witnesses numerous institutional obstacles [7–10], while the most obvious problem of systematic and structural contradictions lies in the aspect of administrative division [11–14]. Strengthening the relevant research on China's administrative regionalization will help the world to better understand the internal mechanism of China's urbanization and provide reference for relevant theoretical and practical research.

In western countries with highly mature market economy, the economic functions of local governments are relatively limited. In the context of the federal decentralization system, the central government has relatively little intervention in the adjustment of administrative divisions. Administrative divisions are mostly formed naturally with the economic and social development, and rarely are adjusted by administrative forces. As the urbanization process has reached a high level, the administrative division of metropolitan areas has become the focus of relevant research, which is also involved in the study of local government related issues [15–24]. Metropolitan area reform has been a hot topic in Europe and The United States for nearly a century due to the practice of metropolitan area governments such as greater London Council of Britain, Metropolitan Republic of France, Metropolitan area government of Canada, and designated city by decree of Japan [25–27]. China's administrative system is quite different from that of other countries. After the reform and opening up, RDC has gradually become an important way to promote urbanization in China. RDC is a research hotspot in recent years, but the academic circle has not reached a consensus on the economic performance and comprehensive effect of county (county-level city) districting [28–31].

Due to the different development stages, different regions may have great differences on the same policy. Therefore, this paper analyzes the factors influencing the adjustment of administrative division when municipal districts are established after the removal of counties (county-level cities) from the aspect of city characteristics. (1) Characteristics of city scale. Compared with medium and small cities, big cities, with the permanent resident population of more than 1 million in the urban area, achieve relatively mature development and have relatively strong radiation and driving effect. In addition, in general, big cities have more registered population, and have stronger attraction to the floating population, thus being part of the focus of urbanization. It is beneficial to promoting the cultivation and development of metropolitan areas and enhancing the radiation and driving role of big cities in medium and small cities by straightening out the administrative system of municipal districts in big cities. (2) Characteristics of city location. The development stages of cities tend to be greatly different due to their different locations. Since the reform and opening up, the levels of industrialization and urbanization in the eastern region are elevated rapidly, and the administrative division of municipal districts is adjusted much more frequently. A larger number of peripheral counties (county-level cities) in big cities in the eastern region, in particular, are adjusted from 2000 to 2004, while the number of adjusted municipal districts in the central and western regions, especially the common prefecture-level cities, is relatively small. With the implementation of national strategies like the "Belt and Road" initiative, however, the central and western regions are facing great development opportunities of industrialization and urbanization, and are also in urgent actual need of the adjustment of municipal districts, while the administrative division of municipal districts in the northeastern region has their own uniqueness because the large-scale industrialization period often happens before or at the early stage of reform. (3) Development intensity of municipal districts. The municipal districts, as the component unit of inner cities, are established as required by the development of metropolitan areas and urban administration after cities have developed to a certain scale. If the central urban area has urgent actual need, the investment made in the infrastructure, public services, project layout, etc. in newly established

districts will be increased, thus it is easier to enhance the overall economic effectiveness of such region. The following hypotheses are hereby proposed:

Hypothesis 1: Municipal district for county (county-level city) is relevant to city scale, and compared with the medium and small cities, the establishment of municipal districts in big cities after the removal of counties (county-level cities) has stronger economic growth effect.

Hypothesis 2: Municipal district for county (county-level city) is relevant to locational conditions, and the effect of municipal district for county (county-level city) in the eastern and central regions is more obvious.

Hypothesis 3: The greater the development intensity of central municipal districts, the more urgent the actual demand for the municipal district for county (county-level city).

## Positivism model and data source

### Positivism model

The adjustment of administrative division in municipal districts shall be approved by China State Council with no definite standard of establishment for a long time and with a long period of application, thus the economic growth effect of district for county (county-level city) has some certain advantages of "quasi natural experiment". After referring to the relevant research literature on performance analysis [32–39], this paper adopts the Difference-in-Difference (DID) model commonly used for policy evaluation to evaluate the policy effect made by the adjustment of administrative division of municipal district for county (county-level city). DID comes from the aggregate data model of econometrics, which is a widely used econometric method in policy analysis and engineering evaluation. It is mainly applied to evaluate the impact degree of a certain event or policy in mixed cross section data set. The dual difference method does not require the experimental group and the control group to be completely consistent, there may be some differences between the two groups, allowing selection according to individual characteristics, as long as the characteristics do not change with time, which is the biggest advantage of DID. That is to partially alleviate the "selection bias" -induced endomutation. Therefore, this study chooses DID method to study RDC. The main idea of DID model lies in that the two aspects of difference, horizontal unit and time series, brought about by the exogenous public policies are counted to finally identify the "treatment effect" of public policies [40]. Among the city samples nationwide, the cities (counties) removed for municipal districts from 1998 to 2016 are regarded as the treatment group, and the cities not adjusted from 1998 to 2016 are regarded as the control group. This paper, by referring to relevant research results, sets the DID standard econometric model as follows:

$$y_{it} = \alpha + \beta\, Policy_{it} + \theta X_{it} + u_i + v_i + \varepsilon_i \tag{1}$$

Where represents the economic performance of city i in the year of t, the core dependent variable is per capita real GDP; represents the dummy variable of policy implementation, the cities after the adjustment of municipal districts from 1998 to 2016 are the treatment group, with the value of this variable as 1, and the cities not adjusted are the control group, with the value of this variable as 0; represents the control variable, the following control variables are chosen by referring to the current researches [41–43], specifically including: urbanization level (urban), which is the ratio of non-agricultural population to total population; development intensity of municipal districts (inten), which is the ratio of area of construction land in the municipal

districts to the area of the municipal districts; population density of municipal districts (pden), which is the ratio of the population of municipal districts to the area of municipal districts; human capital (edu), which is the number of on-campus students in secondary and higher institutions; level of foreign direct investment (fdi), which is the ratio of disbursement of foreign capital to GDP; government scale (gov), which is the ratio of expenditures within the government budget to GDP; β represents the core regression parameter, which reflects the policy effect of the adjustment of municipal districts; represents the time fixed effect, represents the individual fixed effect, and is the random disturbance term. Adopting natural logarithm is conducted for all the basic data for the purpose of eliminating the influence of different dimensions on the results.

Over the twenty years before the reform and opening up, municipal districts were added by means of prefecture upgraded to city, prefecture and city merger and so on. After the suspension of approval for county upgraded to city in 1997, the adjustment of administrative division in municipal districts began to enter the "period of big city dominance" from the "period of prefecture and city merger". In addition, the revenue and expenditure logic of local finance has witnessed obvious changes after the reform of tax distribution system in 1994. The reasons for using 1998–2016 as the research period mainly include: first, in 1997, China suspended the work of withdrawing counties and establishing cities, and the number of withdrawing counties and cities began to increase greatly; Second, after 2017, the withdrawal of counties (cities) into districts began to be gradually tightened, and gradually entered the "stage of strict control"; Third, since 2019, due to the impact of COVID-19, the economic situation has been stagnant, which has had a certain impact on the adjustment effect of administrative divisions; Fourth, referring to relevant literature [28, 29], because the adjustment effect appears with a certain lag, most of the data in the past period of time is not used for analysis. So the research phase of this paper is thus identified as the cities undergoing the adjustment of administrative division of municipal districts nationwide from 1998 to 2016, which is the period when municipal district for county (county-level city) happened frequently. Data at the level of municipal districts are used in this paper, and the population, GDP, financial revenue and expenditure and other relevant data of the removed and merged counties before their removal and merger are added to the cities, to eliminate the estimation deviation caused solely by the adjustment of administrative division.

## Data source

Data relevant to the adjustment of administrative division are derived from the China Administrative Division Network (http://www.xzqh.org/html/). The economic data of cities come from The Statistical Yearbook of Chinese Cities and Statistical Yearbook of Chinese Urban Construction provided by China Economic and Social Big Data Research Platform(https://data.cnki.net/Yearbook/Navi?type=type&code=A) in past years, and the data of the counties (cities) removed from the merger come from the Statistical Yearbook of Chinese Counties and the statistical yearbook of provinces and cities in past years. Based on the data collection and collation, the samples are preprocessed, specifically including: first, to exclude the cities with a large number of missing data, and use the interpolation method to supplement the specific missing and abnormal data; second, to exclude the data of municipalities directly under the Central Government, because the management system of municipalities directly under the Central Government has certain particularity; third, the nominal variables like GDP are adjusted to the real variables taking year 1998 as the base period through the consumer price index (CPI) of each city. A total of 240 cities are sampled in the final research. In consideration

of index lack in some specific cities in the statistical yearbook, the unbalanced panel data(consists of the attachment 1 for this submission) are adopted in this research.

## Results and discussion

This research evaluates the economic effect of municipal district for county (county-level city) mainly from three aspects: at first, to analyze the Average Treatment Effect (ATE) of the adjustment of municipal districts on urban economic growth; then, to analyze the heterogeneity of policy effect; finally, to carry out the robustness test.

### Baseline regression results

The table reports the average treatment effect of the adjustment of administrative division of municipal districts on urban economic growth based on the equation setting of Model (1). In order to solve the possible problems relevant to time series and heteroscedasticity in the DID estimation with a longer time span, the relatively strict clustering standard error at the city level is adopted for the regression. As can be seen from the Table 1, the regression coefficients of the difference-in-difference estimators of the municipal district for county (county-level city) are all significantly positive. Line (1), the estimation results without control variables, is positive at the significance level of 10%. After adding the development intensity, investment rate, government scale and other control variables, and controlling time and city fixed effect, the estimation coefficient increases to 0.124, and is significantly positive at the level of 1%, which means that the municipal district for county (county-level city) has a certain promoting

**Table 1. The estimation results of the economic growth effect of municipal district for county (county-level city).**

| variable | lnpgdp | lnpgdp |
|---|---|---|
| | (1) | (2) |
| Policy | 0.050 | 0.117*** |
| | (1.56) | (3.73) |
| urban | | 0.175*** |
| | | (4.11) |
| inten | | 0.088*** |
| | | (4.35) |
| pden | | -0.055 |
| | | (-1.43) |
| edu | | -0.043 |
| | | (-1.43) |
| fdi | | 0.003 |
| | | (0.54) |
| gov | | -0.214*** |
| | | (-7.17) |
| Constant | 9.014*** | 9.909*** |
| | (501.10) | (38.34) |
| Year FE | Yes | Yes |
| City FE | Yes | Yes |
| N | 4512 | 3984 |
| $R^2$ | 0.949 | 0.961 |

Note: The content reported in the brackets is t statistic, and ***, ** and * respectively represent that 1%, 5% and 10% are statistically significant. (the same as tables below).

effect on the urban economic growth. The analysis results are consistent with previous studies [29, 42]. Among the control variables, the level of urbanization and development intensity of municipal districts have significantly promoted the progress of establishing municipal districts after the removal of counties (county-level cities), and the increase of nonagricultural population and extension of construction lands have intensified the expansion of scope of city spaces.

## Analysis of heterogeneity

The basic model reflects the overall influence of municipal district for county (county-level city) on the urban economic growth. Due to the unbalancedness of city development in China since the reform and opening up, cities in the eastern region and big cities are obviously superior to other cities, then the author wonders if the adjustment of municipal districts will make similar effect on cities in different regions or cities with different scales. In order to test the different influences of distinctive adjustment methods on urban economy, this paper, based on the basic regression model, chooses three classification standards, including city scale, city location and development intensity of municipal districts, to carry out regression according to sample classification, for the purposes of measuring the heterogeneity of policy effect. Among the three standards, city scale is classified according to the permanent resident population in urban area in 2016.

This interaction term of city scale and adjustment policy (Policy×scale) is added to this research based on the basic model. The difference of estimation effects on cities with different characteristics is tested by the coefficients of interaction terms. If the coefficient of interaction term is significant, it suggests that there is obvious heterogeneity. As shown in the Table 2, after the city scale is classified as the medium and small cities with the permanent resident

**Table 2. Heterogeneity analysis of policy effect (city scale).**

| varile | Less than 1 million | 1–5 million | More than 5 million |
|---|---|---|---|
| Policy | 2.551*** | 0.243 | 1.133** |
|  | (4.16) | (0.42) | (2.21) |
| Policy×scale | -2.597*** | 2.266*** | 2.228*** |
|  | (-3.13) | (2.66) | (3.14) |
| urban | -0.629 | -0.654 | -0.684* |
|  | (-1.55) | (-1.61) | (-1.67) |
| inten | 0.777*** | 0.811*** | 0.792*** |
|  | (2.78) | (2.92) | (2.59) |
| pden | 0.318 | 0.296 | 0.328 |
|  | (0.51) | (0.48) | (0.52) |
| edu | 0.940*** | 0.932*** | 0.929*** |
|  | (2.77) | (2.75) | (2.73) |
| fdi | -0.244*** | -0.245*** | -0.252*** |
|  | (-2.86) | (-2.87) | (-2.94) |
| gov | -2.301*** | -2.306*** | -0.283*** |
|  | (-5.44) | (-5.42) | (-5.38) |
| Constant | 0.257 | 0.257 | 0.081 |
|  | (0.06) | (0.06) | (0.02) |
| Year FE | Yes | Yes | Yes |
| City FE | Yes | Yes | Yes |
| N | 3984 | 3984 | 3984 |
| R$^2$ | 0.708 | 0.707 | 0.705 |

population in urban area of less than 1 million, big cities with the population of 1 to 5 million, and super cities with the population of more than 5 million, the coefficients of interaction terms of those three classes of cities are all significant under the level of 1%, among which the regression estimation coefficient of cities with 1 to 5 million people is 2.266, the regression coefficient of cities with more than 5 million people is 2.228, and both of them are significantly positive. Compared with super cities, the effect of municipal district for county (county-level city) in big cities is better, which means that the result is in consistency with the expectation of Hypothesis 1. However, the regression estimation coefficient of the medium and small cities with less than 1 million people is -2.597, which presents significant negative correlation, and shows that the municipal district for county (county-level city) in medium and small cities with relatively few permanent resident population in urban area has obvious negative influence on the economic development. This result is relevant to the cities'relatively limited economic strength, lower autonomy after the county (county-level city) changed into municipal districts, relatively weak radiation and driving effect from the central cities, and difficulty in driving the rise of overall economic level.

In the regional regression models (Table 3), the method that adds the interaction terms is not adopted to conduct the estimation. Owing to the influence of city scale and other similar factors, difference of the policy effect among different regions becomes complicated, thus the sample classification is adopted. Results show that the effect of municipal district for county (county-level city) in the eastern and central regions is obviously superior to that of other regions, and is significant at the levels of 5% and 1% respectively, which means that the municipal district for county (county-level city) has obvious positive promoting effect on the economic growth in the eastern and central regions, and the policy effect on the western and northeastern regions is not significant. The result is in consistency with the expectation of Hypothesis 2. Since 1998, the number of municipal district for county (county-level city) is the

**Table 3. Heterogeneity analysis of policy effect (city region).**

| varile | Eastern China | Central China | Western China | Northeast China |
|---|---|---|---|---|
| Policy | 0.109** | 0.250*** | 0.001 | 0.156 |
|  | (2.37) | (3.41) | (0.02) | (1.40) |
| urban | 0.150*** | 0.165* | 0.071 | 0.214 |
|  | (3.20) | (1.96) | (1.22) | (0.78) |
| inten | 0.048 | 0.148*** | 0.052 | 0.070 |
|  | (1.66) | (3.71) | (1.21) | (0.98) |
| pden | -0.103 | -0.045 | 0.027 | -0.223 |
|  | (-1.65) | (-0.73) | (0.33) | (-1.31) |
| edu | 0.109** | 0.250*** | 0.001 | 0.156 |
|  | (2.37) | (3.41) | (0.02) | (1.40) |
| fdi | 0.150*** | 0.165* | 0.071 | 0.214 |
|  | (3.20) | (1.96) | (1.22) | (0.78) |
| gov | 0.048 | 0.148*** | 0.052 | 0.070 |
|  | (1.66) | (3.71) | (1.21) | (0.98) |
| Constant | -0.103 | -0.045 | 0.027 | -0.223 |
|  | (-1.65) | (-0.73) | (0.33) | (-1.31) |
| Year FE | Yes | Yes | Yes | Yes |
| City FE | Yes | Yes | Yes | Yes |
| N | 1180 | 1502 | 732 | 570 |
| $R^2$ | 0.968 | 0.966 | 0.971 | 0.955 |

most in the eastern region, and mainly concentrates on Jiangsu, Guangdong and Zhejiang provinces, especially from 2000 to 2004. It effectively meets the actual demand of the economic development in the Pearl River Delta and Yangtze River Delta. In addition, the rapid industrialization and urbanization of the eastern coastal areas over the recent 20 years also promotes the adequate development of newly established municipal districts. The reform of municipal district for county (county-level city) in the central region has been intensive since the 18th National Congress of the CPC. The major effects include: first, it effectively eases one district in one city, the county surrounding city, the cities and counties in the same urban region and other problems in some cities, such as the municipal district for county in Xuchang, Kaifeng and other cities; second, it effectively meets the actual demands of industrialization and urbanization in the central region. As most coastal developed cities have entered the post-industrialization stage, "vacating cage to change bird" and industry transformation and upgrading have emerged as the key points of economic development. However, at present, the central region is at the critical stage of industrialization and urbanization, and the population, resources, and other factors need to flow back from the eastern region on a large-scale basis, which promotes the actual demand of the central region for adding the municipal districts correspondingly. In recent years, the municipal district for county (county-level city) in central cities such as Changsha, Nanchang and so on has important significance on its province. An indepth thinking shall also be conducted for the nonsignificant policy effect of the western and northeastern regions. Since 1998, the number of municipal district for county (county-level city) is relatively small in those two regions, and mainly concentrates on the central cities such as Chongqing, Chengdu, Harbin, Shenyang, Changchun, Dalian, etc. On the one hand, such situation happens due to the stage characteristics of regional development, and the insufficient actual demand of common prefecture-level cities; on the other hand, such situation also causes the increasingly large area and remote distance and other problems of municipal districts in Harbin and other similar cities. For example, Liaozhong District of Shenyang, Jiutai District of Changchun, Shuangcheng District of Harbin, Pulandian District of Dalian are more than 50km away from the government offices of central cities. Those factors, to varying degrees, have resulted in the problem that the policy effect of municipal district for county (county-level city) is not significant. Therefore, during the process of municipal district for county (county-level city), the summary of successful experience in eastern and central regions shall be paid more attention to. What's more, the differentiated classification guidance shall be given in combination with the development stage and actual demand of each region.

The regression result in the Table 4 shows that, if the development intensity of municipal districts is different, there will be a big difference in the economic effect of municipal district for county (county-level city); if the development intensity of municipal districts is greater than 10%, the estimation coefficient is 0.170; under the condition of 1%, the result is significantly positive; if it is less than 10%, the regression result is not significant. However, if the development intensity of municipal districts is greater than 15%, the policy effect increases to 0.257, and is significant under the condition of 1%, which is in consistency with the expectation of Hypothesis 3. This shows that, when the municipal district for county (county-level city) is propelled, the development intensity of municipal districts in central cities must be paid attention to; when the development space of municipal districts is limited, better effect of municipal district for county (county-level city) is easy to be achieved. However, it does not mean that bigger development intensity of municipal districts represents better result; when the development intensity of municipal districts is greater than 20%, the estimation coefficient changes not significantly, because the unnecessary system costs are possible when the development intensity of municipal districts is too high, and it is not beneficial to the longterm

**Table 4. Heterogeneity analysis of policy effect (development intensity of municipal districts).**

| varile | inten≥10% | inten<10% | inten≥15% | inten<15% |
|---|---|---|---|---|
| Policy | 0.170*** | 0.007 | 0.257*** | 0.006 |
| | (3.08) | (0.19) | (3.61) | (0.17) |
| urban | 0.046 | 0.211*** | 0.146 | 0.210*** |
| | (0.46) | (5.85) | (1.12) | (5.83) |
| pden | -0.093 | 0.087 | -0.090 | 0.058 |
| | (-1.10) | (1.43) | (-0.85) | (0.95) |
| edu | 0.029 | -0.035 | 0.085* | -0.053 |
| | (0.60) | (-0.90) | (1.91) | (-1.53) |
| fdi | 0.008 | -0.004 | 0.001 | 0.002 |
| | (0.60) | (-0.64) | (0.06) | (0.36) |
| gov | -0.189*** | -0.226*** | -0.119** | -0.228*** |
| | (-3.93) | (-5.86) | (-2.08) | (-6.61) |
| Constant | 10.360*** | 9.056*** | 10.222*** | 9.321*** |
| | (16.08) | (23.10) | (12.82) | (23.12) |
| Year FE | Yes | Yes | Yes | Yes |
| City FE | Yes | Yes | Yes | Yes |
| N | 1297 | 2691 | 791 | 3197 |
| $R^2$ | 0.955 | 0.962 | 0.949 | 0.963 |

development of cities. This shows that the adjustment of administrative division in municipal districts has a certain window phase of optimal time.

When the city is in a state of rapid development, urban renewal speed will be accelerated, the existing city capital (such as infrastructure) depreciation is forced to shorten time, leading to urban renovation costs (switching costs) has increased dramatically, and fast for the development of the economy and municipal district area of the city is too small, will increase speed of development cost. Zhengzhou in Henan Province, for example, covers an area of only 1010 square kilometers, and its development intensity is close to 50%. As the development space of the central urban area is becoming increasingly tight, which is not good for the spatial layout of industries and public services, the high density of facilities leads to the problem that many newly built facilities need to be removed in the urban renewal process, such as short-lived bus stations, short-lived overpasses and short-lived public toilets. Another example is Harbin, which in recent years has promoted a large number of county (city) reform to establish districts, resulting in a municipal area of more than 10,000 square kilometers. Due to the urban management system of municipal districts, there is a large number of agricultural population, and counties and county-level cities can enjoy poverty alleviation and agricultural benefit policies such as subsidies for livestock breeding and agricultural machinery purchase, but municipal districts cannot enjoy them. This makes such municipal district public service levels cannot be compared with the central city, economic development and not equal to counties and county-level cities, even in accordance with the actual demand cannot enjoy benefit farming policy, one, two, three industry development have encountered difficulties, caused by a slowing economy, low level of urban construction and population agglomeration problems such as weak, This is also an important reason for Harbin's economic slowdown in recent years. This shows that the development intensity of municipal districts should be taken into full consideration when promoting the withdrawal of counties (cities) into districts, as either too early or too late is not conducive to the improvement of economic and social benefits.

## Robustness test

Firstly, parallel trend test. If the DID method is used to evaluate the policy effect, the treatment group and the control group should be ensured to meet the parallel trend before the municipal district for county (county-level city), so as to make sure of the credibility and objectivity of the results. This paper will, based on the method of counterfactual test, evaluate the changes of core dependent variables (per capita real GDP) under the following two conditions (after the RCD and without RCD). According to whether to carry out municipal district for county (county-level city) or not, the samples are divided into two groups: the treatment group with municipal district for county (county-level city) and the control group not affected by the policy of municipal district for county (county-level city). If the economic growth trends of the treatment group and the control group are not significant before the county (county-level city) changed into municipal district, it means that the urban economic growth of the treatment group and the control group has the parallel trend; while if there is significant difference after the policy is implemented, the actual effect of the policy of municipal district for county (county-level city) can be obtained. In this paper, the actual time of municipal district for county (county-level city) is advanced by two years, and two dummy variables, one year before the municipal district for county (county-level city)$(Policy_{i,-1})$ and two years before the municipal district for county (county-level city)$(Policy_{i,-2})$, and the variables of treatment group are added to carry out the test. If the policy of municipal district for county (county-level city) has no significant influence on the urban economic growth in one year and two years before the policy, it shows that the result meets the hypothesis of parallel trend. As shown in Table 5, the dummy variables, one year and two years before the municipal district for county (county-level city), are not significant, which means that the influence of the policy

**Table 5. Policy effect of municipal district for county (county-level city): Counterfactual test.**

| variable | Withdrawal of counties (county-level cities) one years before the establishment of municipal districts | Withdrawal of counties (county-level cities) two years before the establishment of municipal districts |
|---|---|---|
| Policy | 0.326 | -0.241 |
|  | (0.47) | (-0.49) |
| urban | -0.518 | -0.441 |
|  | (-1.26) | (-1.06) |
| inten | 0.559* | 0.452 |
|  | (1.94) | (1.60) |
| pden | 0.560 | 0.689 |
|  | (0.89) | (1.11) |
| edu | 1.039*** | 1.089*** |
|  | (3.06) | (3.19) |
| fdi | -0.244*** | -0.243*** |
|  | (-2.82) | (-2.81) |
| gov | -2.286*** | -2.277*** |
|  | (-5.40) | (-5.40) |
| Constant | -1.100 | -1.802 |
|  | (-0.26) | (-0.43) |
| Year FE | Yes | Yes |
| City FE | Yes | Yes |
| N | 3984 | 3984 |
| $R^2$ | 0.703 | 0.703 |

**Table 6. Results of robustness test.**

| variable | Change the control variable | Full sample regression | Change the explained variable |
|---|---|---|---|
| | (1) | (2) | (3) |
| Explained variable | Real GDP per capita | Real GDP per capita | Night light index |
| Policy | 0.169*** | 0.113*** | 0.083*** |
| | (4.91) | (3.67) | (2.68) |
| urban | | 0.176*** | 0.047 |
| | | (4.21) | (1.54) |
| inten | 0.101*** | 0.087*** | 0.052*** |
| | (-6.87) | (4.39) | (3.14) |
| pden | -0.040 | -0.053 | -0.064** |
| | (-0.97) | (-1.38) | (-2.20) |
| edu | -0.382 | -0.044 | -0.009 |
| | (-1.24) | (-1.46) | (-0.47) |
| fdi | 0.002 | 0.003 | 0.008** |
| | (0.34) | (0.46) | (1.98) |
| gov | -0.208*** | -0.211*** | 0.014 |
| | (-6.87) | (-7.23) | (0.82) |
| Constant | 9.635*** | 9.905*** | 2.339*** |
| | (35.98) | (38.57) | (12.40) |
| Year FE | Yes | Yes | Yes |
| City FE | Yes | Yes | Yes |
| N | 3985 | 4091 | 2979 |
| $R^2$ | 0.960 | 0.962 | 0.828 |

on the treatment group and the control group is unbiased in one year and two years before the policy of municipal district for county (county-level city). After the county (county-level city) changed into municipal district, the interaction terms of municipal district for county (county-level city) and the treatment variable are significantly positive, that is to say, the municipal district for county (county-level city) has significant influence on the urban economic growth.

Secondly, change the control variables. Since the non-agricultural population and nonagricultural economy gather in cities, whether the municipal districts, as the urban administrative districts, have typical city characteristics is closely related to the economic development. Therefore, this research carries out the reestimation for the basic regression model after excluding the proportion variable of nonagricultural population. As shown in Line (1) of the Table 6, the value of regression result and the level of significance are in consistency with results in the Table 1, which further proves the robustness of the basic regression results. In addition, in the basic regression, in consideration of the certain particularity of the management of municipalities directly under the Central Government, the data relevant to municipalities directly under the Central Government are excluded to ensure the accuracy and objectivity of the results. The excluded municipalities directly under the Central Government are added to the full-sample regression to test the robustness of basic regression results. The results are shown in Line (2) of the Table 6, from which it can be seen that the regression results still keep the 1% level of significance, which is in consistency with the basic regression results.

Thirdly, test of light index. The night light data can reflect the conditions of economic activities on a more objective basis [44–48]. The intensity images of night light can be utilized to

build the comprehensive light index to reflect the level of urbanization, to effectively avoid the problem of over estimation of official economic statistical data, and to achieve more objective and true estimation with higher continuity of data. This paper adopts the Global Night Light Data (DMSP) from 2000 to 2013 released by the Resource and Environmental Science and Data Center of Chinese Academy of Sciences to reflect the economic development level of municipal districts. The value of light brightness ranges from 0 to 63; the bigger the value, the brighter the light, showing the higher the comprehensive development level of this region. The contrast between the bright and dark areas of the light images makes it a good data source of research on the urban economy. In this paper, the boundary base map of the administrative division in the municipal districts of cities in 2016 is superimposed on the night light grid data, which is equal to the direct addition of the data of removed and merged county (county-level city) to the municipal districts at the same time. The light brightness index of each city is obtained by dividing the sum of all brightness values of grid light within the range of its municipal districts by the total number of grids. As shown in Line (3) of the Table 6, if the per capita real GDP is replaced by the night light index for reestimation, the result is still significant at the level of 1%, and there is no great difference among the estimation coefficients, which further proves the robustness of policy estimation effect.

## Conclusion and discussion

In recent two decades, the development of industrialization in China has gradually transformed from the light industry driving mode before 1998 that emphasizes on meeting the domestic market demand to the heavy industry driving mode that provides services for the global market demand; the development of urbanization has gradually transformed from increasing the number of cities by means of municipal districts changed into prefecture-level cities and counties removed for cities before 1998 to expanding the city scale by means of municipal district for county (county-level city) with the rapid development of the real estate market. However, with the large-scale expansion of cities, the problem that the land urbanization is faster than the population urbanization is increasingly prominent. The author wonders whether the policy of municipal district for county (county-level city) significantly promotes the urban economic growth or not. The DID model is adopted in this paper to conduct empirical analysis of this problem. The research shows that:

### Conclusion

1. In recent years, China's policy of municipal district for county (county-level city) significantly promotes the urban economic growth. In the basic regression model, the regression estimation coefficient is significantly positive, and maintains stable after changing the variables and conducting the parallel trend test.

2. In the aspect of city scale, the effects of municipal district for county (county-level city) in big and super cities are obviously superior to those of the medium and small cities. Therefore, during the process of municipal district for county (county-level city) in the future, close attention shall be paid to the big cities with the permanent resident population of more than 1 million in urban area, the radiation and driving ability of big cities shall be enhanced, and the development vitality of medium and small cities shall be further activated effectively; as for the adjustment of medium and small cities, not only the development demand of central urban areas, but also the urbanization levels of the removed and merged counties (county-level cities) shall be considered, and the methods of adjustment shall be more flexible.

3. In terms of the region location, the effect of county (county-level city) changed into municipal district in the eastern and central regions is more significant. In recent two decades, the number of adjustment is the most in the eastern region, and favorable effects have been achieved. This period has also witnessed the rapid development of industrialization and urbanization in the eastern region, which shows that the scale expansion of municipal districts must be suited to the stage of economic development. In recent years, the rapid rise of central region is closely related to the increased development pressure and industrial dispersion of the eastern region. At present, the central and western regions are facing the large-scale back flow of population and industries, and are also at the critical stage of industrialization and urbanization, thus more attention shall be paid to the problem of undue scale of municipal districts in the cities of the central and western regions.

4. In the aspect of the development intensity of municipal districts, if the development intensity of municipal districts ranges from 15% to 20%, the effect of municipal district for county (county-level city) is better, which is in consistency with the experience of Western developed countries like Japan. Too early municipal district change will result in problems like non-strong driving ability, and will also affect the normal economic development of the removed and merged counties (county-level cities); while too late municipal district change will affect the development of central urban areas. Relevant planning will be difficult to be more scientific and efficient; problems including resource waste will be caused, and the normal development of the whole region will be further affected.

## Discussion

The adjustment of administrative divisions of municipal districts is a multi-disciplinary systematic project with dynamic, complex and comprehensive characteristics. It needs to optimize the layout according to the actual development needs of the state and local areas on the basis of following the differences and changing trends of the spatial distribution of economic factors.

China's experience shows that in the process of rapid urbanization development, first, we should attach great importance to the important significance of administrative division adjustment to the governance of metropolitan areas. Timely promotion of administrative division adjustment is beneficial to reduce the cost of urbanization development and improve the governance efficiency of metropolitan areas, which is also an important experience of rapid urbanization development in the past 40 years of reform and opening up. The rest of the world should learn from this experience in light of their national conditions.

The second is to promote the spatial optimization of municipal districts based on the thinking of urban agglomeration. Scientific and reasonable setting of municipal districts can optimize the scale and structure of urban agglomerations, effectively release the potential of regional spatial development and improve the development efficiency of urban system. The scale and structure of municipal districts should be optimized timely.

Third, the adjustment of municipal districts should focus on the development intensity and development potential of cities. It is necessary to determine whether the central urban area has urgent practical needs. Only when the development of the central urban area is restricted (under normal circumstances, the development intensity of the municipal district is above 15%) and the development intensity and development potential are both large, can it be necessary to set up new municipal districts and expand the area of the municipal district. Promoting

the adjustment of municipal districts at the right time will help alleviate the problems of large cities and provide new ideas for the governance of metropolitan areas.

## Supporting information

**S1 File.**
(XLSX)

## Acknowledgments

Thanks to Dr. Chen Zhuo for his help in data curation, supervision, methodology and other aspects.

## Author Contributions

**Conceptualization:** Biao Zhao.

**Data curation:** Biao Zhao, Xu Xi.

**Formal analysis:** Biao Zhao.

**Funding acquisition:** Biao Zhao.

**Investigation:** Biao Zhao.

**Methodology:** Biao Zhao.

**Project administration:** Biao Zhao.

**Resources:** Biao Zhao.

**Software:** Biao Zhao, Xu Xi.

**Supervision:** Biao Zhao, Xu Xi.

**Validation:** Biao Zhao.

**Visualization:** Biao Zhao.

**Writing – original draft:** Biao Zhao.

**Writing – review & editing:** Biao Zhao.

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
