## [Decision Letter · Decision Letter 0]

19 May 2022

PONE-D-22-10605System transformation and city development: research on economic growth performance of district for county (county-level cities) in ChinaPLOS ONE

Dear Dr. biao zhao,

Thank you for submitting your manuscript to PLOS ONE. After careful consideration, we feel that it has merit but does not fully meet PLOS ONE’s publication criteria as it currently stands. Therefore, we invite you to submit a revised version of the manuscript that addresses the points raised during the review process.

We look forward to receiving your revised manuscript.

Kind regards,

Carlos Alberto Zúniga-González, Ph.D

Academic Editor

PLOS ONE

Journal Requirements:

4. We note you have included a table to which you do not refer in the text of your manuscript. Please ensure that you refer to Tables 1, 2, 3, 4 and 6 in your text; if accepted, production will need this reference to link the reader to the Table.

Additional Editor Comments:

Dear author for the shake the improve of your manuscript I suggest to follow the comment of the reviewers. The manuscript has potentials, but is necessary to make improvement. Is necessary to justify why the data set is old. With regard to references and citations, they do not conform to the journal's criteria. The references must be numbered [1] and in the citations put the number. I note that it has 16 references and that is very poor for the type of article they write. I suggest adding more quotes at least 40.

I sugesst the folowing references:

[1] González, C. A. Z. (2011). Technical efficiency of organic fertilizer in small farms of Nicaragua: 1998-2005. African Journal of Business Management, 5(3), 967-973. Available from publons.com/p/11272633/

[2] Dios-Palomares, R. (2015). 7. Analysis of the Efficiency of Farming Systems in Latin America and the Caribbean Considering Environmental Issues. Revista Cientifica-Facultad de Ciencias Veterinarias, 25(1). Available in publons.com/p/3106827/

[3] Zuniga González, C. A. (2020). Total factor productivity growth in agriculture: Malmquist index analysis of 14 countries, 1979-2008. Revista Electrónica De Investigación En Ciencias Económicas, 8(16), 68–97. https://doi.org/10.5377/reice.v8i16.10661

[4] Dios-Palomares, R., Lopez de Pablo, D., Diz Pérez, J., Jurado Bello, M., Guijarro, A., Martinez-Paz, J., & Zúniga González, C. (2015). Environmental aspects in the analysis of efficiency. Rev. Iberoam. Bioecon. Cambio Clim., 1(1), 88-95. https://doi.org/10.5377/ribcc.v1i1.2143

[5] López-González, Álvaro, Zúniga-González, C., López, M., Quirós-Madrigal, O., Colón-García, A., Navas-Calderón, J., Martínez-Andrades, E., & Rangel-Cura, R. (2016). State of the art for measuring productivity and technical efficiency in Latin America: Nicaragua Case. Rev. Iberoam. Bioecon. Cambio Clim., 1(2), 76-100. https://doi.org/10.5377/ribcc.v1i2.2478

[6] Zuniga-Gonzalez, Carlos Alberto (2021), “Total factor productivity in the INTA Chinandega rice variety”, Mendeley Data, V2, doi: 10.17632/76m7p7mvsg.2 https://data.mendeley.com/datasets/76m7p7mvsg/2

[7] Figueroa-Ugalde, J. H., Lagarda-Leyva, E. A., & Celaya-Figueroa, R. (2022). Fundamentos de la sustentabilidad en la bioeconomía y su relación con las teorías administrativas. Rev. Iberoam. Bioecon. Cambio Clim., 8(15), 1806–1821. https://doi.org/10.5377/ribcc.v8i15.14183

[8] Zúniga-González, C. A., López, M. R., Icabaceta, J. L., Vivas-Viachica, E. A., & Blanco-Orozco, N. (2022). Epistemología de la Bioeconomia. Rev. Iberoam. Bioecon. Cambio Clim., 8(15), 1786–1796. https://doi.org/10.5377/ribcc.v8i15.13986

Reviewers' comments:

Reviewer's Responses to Questions

**Comments to the Author**

1. Is the manuscript technically sound, and do the data support the conclusions?

Reviewer #1: Yes

Reviewer #2: Partly

2. Has the statistical analysis been performed appropriately and rigorously? 

Reviewer #1: Yes

Reviewer #2: I Don't Know

3. Have the authors made all data underlying the findings in their manuscript fully available?

Reviewer #1: No

Reviewer #2: No

4. Is the manuscript presented in an intelligible fashion and written in standard English?

Reviewer #1: Yes

Reviewer #2: Yes

5. Review Comments to the Author

Reviewer #1: Having read the manuscript, the manuscript has some potentials that the readership will find insightful but I have some reservations which I will want the author(s) to work on:

1) the INTRODUCTION need to be properly situated by pointing out (i) the justification for the study, (ii) gaps in the literature, (iii) scope justification, (iv) are the outcomes generalisable.

2) more discourse on the empirical approach and technique. Why is this technique used? Is it the best method that addresses the objectives of the study?

3) recast the Policy Recommendations

4) some minor grammatical and editorial corrections are required

Reviewer #2: - the paper does not show concrete policy recommendations

- the dataset is old, not consider relevant factors that may contribute to policy development

- the author did not test for flexible models. True effects to account for the special characteristics of each administrative unit

6. PLOS authors have the option to publish the peer review history of their article (what does this mean?). If published, this will include your full peer review and any attached files.

Reviewer #1: No

Reviewer #2: No

---

## [Author Response · Author response to Decision Letter 0]

13 Jun 2022

Dear Editor:

Thanks to the editorial teachers and external audit experts put forward valuable opinions, so that this paper benefited a lot. According to the comments and suggestions of reviewers, this paper is modified as follows:

Reviewer #1:

1.amendments: “the INTRODUCTION need to be properly situated by pointing out (i) the justification for the study, (ii) gaps in the literature, (iii) scope justification, (iv) are the outcomes generalisable.”

Modification Description: “Thanks for the comments of the experts on this article. The introduction has been modified according to the suggestions of the external audit experts, ensuring that the justification for the study and other contents are included.”

2.amendments: “more discourse on the empirical approach and technique. Why is this technique used? Is it the best method that addresses the objectives of the study?”

Modification Description: It has been modified according to relevant requirements. Positivism Model and Data Source can be seen in detail. In the part of research methods, “DID comes from the aggregate data model of econometrics, which is a widely used econometric method in policy analysis and engineering evaluation. It is mainly applied to evaluate the impact degree of a certain event or policy in mixed cross section data set. The dual difference method does not require the experimental group and the control group to be completely consistent, there may be some differences between the two groups, allowing selection according to individual characteristics, as long as the characteristics do not change with time, which is the biggest advantage of DID. That is to partially alleviate the “selection bias” -induced endomutation. Therefore, this study chooses DID method to study RDC.” In the part of research time, due to the lag in the effect of administrative division adjustment, the RDC gradually entered the stage of strict control after 2017, and China's economy was greatly affected by the SINO-US trade war and the COVID-19 epidemic, the effect of administrative division adjustment was affected to varying degrees. So the research phase of this paper is thus identified as the cities undergoing the adjustment of administrative division of municipal districts nationwide from 1998 to 2016, which is the period when municipal district for county (county-level city) happened frequently. 

3.amendments: “recast the Policy Recommendations”

Modification Description: It has been modified according to relevant requirements. See Conclusion and Discussion for details. It includes the following contents. “The adjustment of administrative divisions of municipal districts is a multi-disciplinary systematic project with dynamic, complex and comprehensive characteristics. It needs to optimize the layout according to the actual development needs of the state and local areas on the basis of following the differences and changing trends of the spatial distribution of economic factors. China's experience shows that in the process of rapid urbanization development, first, we should attach great importance to the important significance of administrative division adjustment to the governance of metropolitan areas. Timely promotion of administrative division adjustment is beneficial to reduce the cost of urbanization development and improve the governance efficiency of metropolitan areas, which is also an important experience of rapid urbanization development in the past 40 years of reform and opening up. The rest of the world should learn from this experience in light of their national conditions. The second is to promote the spatial optimization of municipal districts based on the thinking of urban agglomeration. The scientific setting of urban agglomeration scale structure can effectively release the potential of regional spatial development and improve the development efficiency of urban system. Third, the adjustment of municipal districts should focus on the development intensity and development potential of cities. It is necessary to determine whether the central urban area has urgent practical needs. Only when the development of the central urban area is restricted (under normal circumstances, the development intensity of the municipal district is above 15%) and the development intensity and development potential are both large, can it be necessary to set up new municipal districts and expand the area of the municipal district.” 

4.amendments: “some minor grammatical and editorial corrections are required”

Modification Description: Modifications have been made as required, including title, text, references, formatting, grammatical and punctuation errors, etc.

Reviewer #2:

1.amendments: “the paper does not show concrete policy recommendations”

Modification Description: It has been modified as required.

2.amendments: “the dataset is old, not consider relevant factors that may contribute to policy development”

Modification Description: It has been explained in the paper. The adjustment of administrative division itself is a policy factor. This paper mainly discusses the effect of the policy on economic and social development.

3.amendments: “the author did not test for flexible models. True effects to account for the special characteristics of each administrative unit”

Modification Description: This paper has considered the difference of policy effects under different city sizes, different distribution regions and different development intensities. Other factors will be further detailed and discussed in future studies.

Opinions of the editorial Department:

1.amendments: “Please ensure that your manuscript meets PLOS ONE's style requirements, including those for file naming.”

Modification Description: It has been modified as required.

2.amendments: Questions about financial Disclosure

Modification Description: It has been explained in the article and in the cover letter.

3.amendments: ORCID iD matters

Modification Description: It has been Modified and perfected. https://orcid.org/ 0000-0001-7248-8891

4.amendments: “We note you have included a table to which you do not refer in the text of your manuscript. Please ensure that you refer to Tables 1, 2, 3, 4 and 6 in your text; if accepted, production will need this reference to link the reader to the Table.”

Modification Description: It has been modified in the paper, and each table has been marked in the paper.

5.amendments: Problems with references

Modification Description: It has been revised as required, systematic review of relevant studies has been carried out, and the number of references has been increased to 41.

Finally, I would like to thank the reviewers and editorial department for their valuable comments.

---

## [Decision Letter · Decision Letter 1]

10 Jul 2022

PONE-D-22-10605R1Economic effects of conversion from county (or county-level city) to municipal district in ChinaPLOS ONE

Dear Dr. zhao,

Thank you for submitting your manuscript to PLOS ONE. After careful consideration, we feel that it has merit but does not fully meet PLOS ONE’s publication criteria as it currently stands. Therefore, we invite you to submit a revised version of the manuscript that addresses the points raised during the review process.

We look forward to receiving your revised manuscript.

Kind regards,

Carlos Alberto Zúniga-González, Ph.D

Academic Editor

PLOS ONE

Journal Requirements:

Additional Editor Comments:

Dear, I can see you have improved, however, reviewer 2 insists on Reject, so I need you to make a minor revision and focus it on reviewer 2 observations. Also, I insist on suggesting to you the following references, I think that section methodology can help in your review (minor review). Regarding observation of reviewer 2 I need your minor review and comments.

[1] González, C. A. Z. (2011). Technical efficiency of organic fertilizer in small farms of Nicaragua: 1998-2005. African Journal of Business Management, 5(3), 967-973. Available from publons.com/p/11272633/

[2] Dios-Palomares, R. (2015). 7. Analysis of the Efficiency of Farming Systems in Latin America and the Caribbean Considering Environmental Issues. Revista Cientifica-Facultad de Ciencias Veterinarias, 25(1). Available in publons.com/p/3106827/

[3] Zuniga González, C. A. (2020). Total factor productivity growth in agriculture: Malmquist index analysis of 14 countries, 1979-2008. Revista Electrónica De Investigación En Ciencias Económicas, 8(16), 68–97. https://doi.org/10.5377/reice.v8i16.10661

[4] Dios-Palomares, R., Lopez de Pablo, D., Diz Pérez, J., Jurado Bello, M., Guijarro, A., Martinez-Paz, J., & Zúniga González, C. (2015). Environmental aspects in the analysis of efficiency. Rev. Iberoam. Bioecon. Cambio Clim., 1(1), 88-95. https://doi.org/10.5377/ribcc.v1i1.2143

[5] López-González, Álvaro, Zúniga-González, C., López, M., Quirós-Madrigal, O., Colón-García, A., Navas-Calderón, J., Martínez-Andrades, E., & Rangel-Cura, R. (2016). State of the art for measuring productivity and technical efficiency in Latin America: Nicaragua Case. Rev. Iberoam. Bioecon. Cambio Clim., 1(2), 76-100. https://doi.org/10.5377/ribcc.v1i2.2478

[6] Zuniga-Gonzalez, Carlos Alberto (2021), “Total factor productivity in the INTA Chinandega rice variety”, Mendeley Data, V2, doi: 10.17632/76m7p7mvsg.2 https://data.mendeley.com/datasets/76m7p7mvsg/2

[7] Figueroa-Ugalde, J. H., Lagarda-Leyva, E. A., & Celaya-Figueroa, R. (2022). Foundations of sustainability in the bioeconomy and its relationship with administrative theories. Rev. Iberoam. Bioecon. Cambio Clim., 8(15), 1806–1821. https://doi.org/10.5377/ribcc.v8i15.14183

[8] Zúniga-González, C. A., López, M. R., Icabaceta, J. L., Vivas-Viachica, E. A., & Blanco-Orozco, N. (2022). Bioeconomy Espitemology. Rev. Iberoam. Bioecon. Cambio Clim., 8(15), 1786–1796. https://doi.org/10.5377/ribcc.v8i15.13986

Reviewers' comments:

Reviewer's Responses to Questions

**Comments to the Author**

1. If the authors have adequately addressed your comments raised in a previous round of review and you feel that this manuscript is now acceptable for publication, you may indicate that here to bypass the “Comments to the Author” section, enter your conflict of interest statement in the “Confidential to Editor” section, and submit your "Accept" recommendation.

Reviewer #1: All comments have been addressed

Reviewer #2: (No Response)

2. Is the manuscript technically sound, and do the data support the conclusions?

Reviewer #1: Yes

Reviewer #2: Partly

3. Has the statistical analysis been performed appropriately and rigorously? 

Reviewer #1: Yes

Reviewer #2: I Don't Know

4. Have the authors made all data underlying the findings in their manuscript fully available?

Reviewer #1: Yes

Reviewer #2: Yes

5. Is the manuscript presented in an intelligible fashion and written in standard English?

Reviewer #1: Yes

Reviewer #2: Yes

6. Review Comments to the Author

Reviewer #1: Having read the revised version, I am satisfied with corrections made by the author(s). I find the manuscript clearer to understand with clearly stated objectives and findings. The Policy Recommendations are also relatable to the findings of the study.

Reviewer #2: - The document uses old dataset that results in unclear policy implications

- The paper does not compare the current situation with estimated results

- There is no discussion made with the literature with the results

7. PLOS authors have the option to publish the peer review history of their article (what does this mean?). If published, this will include your full peer review and any attached files.

Reviewer #1: No

Reviewer #2: No

---

## [Author Response · Author response to Decision Letter 1]

13 Jul 2022

Dear Editor:

Thanks to the editorial teachers and external audit experts put forward valuable opinions, so that this paper benefited a lot. According to the comments and suggestions of reviewers, this paper is modified as follows:

Reviewer #1:

amendments: “All comments have been addressed.”

Modification Description: “The article was further proofread and revised.”

Reviewer #2:

1.amendments: “The document uses old dataset that results in unclear policy implications.”

Modification Description: The article has made relevant explanation. The reasons for using 1998-2016 as the research period mainly include: first, in 1997, China suspended the work of withdrawing counties and establishing cities, and the number of withdrawing counties and cities began to increase greatly; Second, after 2017, the withdrawal of counties (cities) into districts began to be gradually tightened, and gradually entered the "stage of strict control"; Third, since 2019, due to the impact of COVID-19, the economic situation has been stagnant, which has had a certain impact on the adjustment effect of administrative divisions; Fourth, referring to relevant literature, because the adjustment effect appears with a certain lag, most of the data in the past period of time is not used for analysis. So the research phase of this paper is thus identified as the cities undergoing the adjustment of administrative division of municipal districts nationwide from 1998 to 2016, which is the period when municipal district for county (county-level city) happened frequently.

2.amendments: “The paper does not compare the current situation with estimated results”

Modification Description: This article has been modified as required. In the empirical research part, the relevant cases of Zhengzhou and Harbin are added to prove the relevant estimation results with the actual situation. “When the city is in a state of rapid development, urban renewal speed will be accelerated, the existing city capital (such as infrastructure) depreciation is forced to shorten time, leading to urban renovation costs (switching costs) has increased dramatically, and fast for the development of the economy and municipal district area of the city is too small, will increase speed of development cost. Zhengzhou in Henan Province, for example, covers an area of only 1010 square kilometers, and its development intensity is close to 50%. As the development space of the central urban area is becoming increasingly tight, which is not good for the spatial layout of industries and public services, the high density of facilities leads to the problem that many newly built facilities need to be removed in the urban renewal process, such as short-lived bus stations, short-lived overpasses and short-lived public toilets. Another example is Harbin, which in recent years has promoted a large number of county (city) reform to establish districts, resulting in a municipal area of more than 10,000 square kilometers. Due to the urban management system of municipal districts, there is a large number of agricultural population, and counties and county-level cities can enjoy poverty alleviation and agricultural benefit policies such as subsidies for livestock breeding and agricultural machinery purchase, but municipal districts cannot enjoy them. This makes such municipal district public service levels cannot be compared with the central city, economic development and not equal to counties and county-level cities, even in accordance with the actual demand cannot enjoy benefit farming policy, one, two, three industry development have encountered difficulties, caused by a slowing economy, low level of urban construction and population agglomeration problems such as weak, This is also an important reason for Harbin's economic slowdown in recent years.This shows that the development intensity of municipal districts should be taken into full consideration when promoting the withdrawal of counties (cities) into districts, as either too early or too late is not conducive to the improvement of economic and social benefits.”

3.amendments: “There is no discussion made with the literature with the results.”

Modification Description: This article has been modified as required. It is mainly reflected in the basic regression results, and relevant literature is added for comparative analysis. “The analysis results are consistent with previous studies[29,42].” In addition, countermeasures and suggestions were modified in this study.

Opinions of the editorial Department:

amendments: “Any changes to the reference list should be mentioned in the rebuttal letter that accompanies your revised manuscript.”

Modification Description: It has been modified as required. Compared with the previous version, 25 references are deleted and 8 references 32-38 are added. 

Finally, I would like to thank the reviewers and editorial department for their valuable comments.

---

## [Editor Report · Decision Letter 2]

18 Jul 2022

Economic effects of conversion from county (or county-level city) to municipal district in China

PONE-D-22-10605R2

Dear Dr. biao zhao,

We’re pleased to inform you that your manuscript has been judged scientifically suitable for publication and will be formally accepted for publication once it meets all outstanding technical requirements.

Kind regards,

Carlos Alberto Zúniga-González, Ph.D

Academic Editor

PLOS ONE

Additional Editor Comments (optional):

Dear authors, Congratulations. I have checked and read your improvements. My decision is accept.
---

## [Editor Report · Acceptance letter]

2 Sep 2022

PONE-D-22-10605R2 

Economic effects of conversion from county (or county-level city) to municipal district in China 

Dear Dr. Zhao:

I'm pleased to inform you that your manuscript has been deemed suitable for publication in PLOS ONE. Congratulations! Your manuscript is now with our production department. 

Kind regards, 

on behalf of

Dr. Prof. Carlos Alberto Zúniga-González 

Academic Editor

PLOS ONE